# Colonisation of hospital surfaces from low- and middle-income countries by extended spectrum β-lactamase- and carbapenemase-producing bacteria

Hospital surfaces can harbour bacterial pathogens, which may disseminate and cause nosocomial infections, contributing towards mortality in low- and middle-income countries (LMICs). During the BARNARDS study, hospital surfaces from neonatal wards were sampled to assess the degree of environmental surface and patient care equipment colonisation by Gram-negative bacteria (GNB) carrying antibiotic resistance genes (ARGs). Here, we perform PCR screening for extended-spectrum β-lactamases ($bla_{CTX-M-15}$) and carbapenemases ($bla_{NDM}$, $bla_{OXA-48}$-like and $bla_{KPC}$), MALDI-TOF MS identification of GNB carrying ARGs, and further analysis by whole genome sequencing of bacterial isolates. We determine presence of consistently dominant clones and their relatedness to strains causing neonatal sepsis. Higher prevalence of carbapenemases is observed in Pakistan, Bangladesh, and Ethiopia, compared to other countries, and are mostly found in surfaces near the sink drain. *Klebsiella pneumoniae, Enterobacter hormaechei, Acinetobacter baumannii, Serratia marcescens* and *Leclercia adecarboxylata* are dominant; ST15 *K. pneumoniae* is identified from the same ward on multiple occasions suggesting clonal persistence within the same environment, and is found to be identical to isolates causing neonatal sepsis in Pakistan over similar time periods. Our data suggests persistence of dominant clones across multiple time points, highlighting the need for assessment of Infection Prevention and Control guidelines.

Environmental surfaces and patient care equipment in hospital settings are among the most critical factors in bacterial horizontal transmission events from high-touch surfaces to patients[1–3], with higher hospital-acquired infection (HAIs) rates in low-income and middle-income countries (LMICs) compared to high-income countries (HICs)[4]. Unless appropriate infection prevention and control (IPC) guidelines are applied and include effective cleaning, disinfection and hygiene practices[4–6], environmental reservoirs of multidrug-resistant (MDR) bacteria can develop and direct contact

with these may contribute to HAIs, such as surgical site infections and bloodstream infections[7–14]. High levels of antimicrobial resistance (AMR) can lead to difficulties in the treatment of MDR infections, associated with high mortality[15–18]. Neonates are particularly at risk because of their underdeveloped immune system, and neonatal infection rates in LMICs are 3–20 times higher than in HICs[19]. Furthermore, AMR bacteria with tolerance to disinfectants may survive on environmental surfaces for months, depending on factors such as humidity, temperature, air ventilation and surface

✉e-mail: maria.nietorosado@biology.ox.ac.uk

characteristics[4,9,20], which will further increase transmission likelihood.

Empirical treatment using cephalosporins or carbapenems in LMIC healthcare settings has influenced the progressive emergence of extended-spectrum β-lactamase- (ESBL-) and carbapenemase-producing bacteria. Antibiotic resistance genes (ARGs) such as $bla_{CTX-M-15}$, and $bla_{NDM}$, $bla_{OXA-48}$-like or $bla_{KPC}$ are often plasmid-borne and/or other mobile genetic element- (MGE) associated, enhancing potential for transmission[8].

The most common ESBL- and carbapenemase-producing Gram-negative bacteria (GNB) in hospitals are Enterobacterales[1,3,8,11,12,21,22]. Few publications have shown data regarding the presence of GNB carrying β-lactamase genes on hospital surfaces in LMICs. Moreover, most of these studies performed in LMICs are single sites focused on specific countries, such as Bangladesh, Pakistan, Ethiopia, or Ghana[1,23–25].

Here, we determine the prevalence and diversity of ESBL- and carbapenemase-carrying bacterial species colonising neonatal wards from six countries within the BARNARDS study (Burden of Antibiotic Resistance in Neonates from Developing Societies). Genetic lineages are assessed to determine whether transmission events occurred and whether they are related to bacteria causing neonatal sepsis in the BARNARDS study.

## Results

### GNB colonisation among countries and hospital sites

In total, 6290 hospital surface swabs (HSS) were processed from ten hospitals across six countries; and 60.7% HSS had GNB growth (Table 1). GNB colonisation varied significantly between the countries and hospital sites, with the highest growth in Bangladesh (92%).

### ARG among countries and hospital sites

Overall, 839/6290 (13.3%) of HSS were positive for $bla_{CTX-M-15}$, 338/6290 (5.4%) for $bla_{NDM}$, and 74/6290 (1.2%) for $bla_{OXA-48}$-like. $bla_{KPC}$ was not detected during this study, therefore, from this point, the term ARGs denotes $bla_{CTX-M-15}$, $bla_{NDM}$ and $bla_{OXA-48}$-like. Concomitant carriage of more than one ARG in the same sample was found in ~1% of samples (Supplementary Table 1).

The prevalence of $bla_{CTX-M-15}$, $bla_{NDM}$ and $bla_{OXA-48}$-like ranged between countries (Fig. 1 and Supplementary Table 2). Noticeable differences between hospitals were observed; for instance, in Bangladesh, almost twice as many $bla_{NDM}$ were detected in BK (17.3%) compared to BC (10.5%), whereas $bla_{OXA-48}$-like was four times more frequent in BC (1.2%). Hospital surfaces in RU had higher $bla_{NDM}$ (2.2%) rates compared to RK, and conversely, $bla_{CTX-M-15}$ and $bla_{OXA-48}$-like were more prevalent (24.4% and 1.1%, respectively) in RK. In Nigeria, the prevalence of all the ARGs was highest in one site (NK).

### GNB colonisation and ARG detection among hospital surfaces

To study the correlation between the presence of ARG and surface type, $n = 4126/6290$ samples contained appropriate metadata following data cleaning (Table 2). The 4126 HSS were collected from 309 different surfaces (Supplementary Data 1) and classified into six categories (listed in Table 2; see methods section "Data cleaning and statistical analysis" for details).

Table 3a summarises the differences in GNB growth across surface categories observing the largest growth near sink drains (68.4%). HSS collected from medical equipment were the least contaminated with Gram-negative bacterial colonisation (45.9%). Furthermore, HSS positive for $bla_{CTX-M-15}$ and $bla_{OXA-48}$-like varied significantly ($P < 0.001$ and $P = 0.004$, respectively) between the six surface categories (Table 3b). Although the presence of $bla_{NDM}$ varied, this was not statistically significant ($P = 0.071$), and the ARG presence was higher in surfaces classified as near the sink drain (Table 3). For detailed ARG prevalence on the different 309 surface types collected and on surface categories among all hospital sites, see Supplementary Data 2 and 3, respectively.

To study the correlation between the presence of ARG and a particular time of the year, $n = 4662/6290$ samples contained appropriate metadata following data cleaning. Temporal analysis revealed that the percentage of $bla_{CTX-M-15}$, $bla_{NDM}$ and $bla_{OXA-48}$-like varied significantly between the seven-time bands across the sampling period from November 2015 until January 2018 ($P < 0.001$ for $bla_{CTX-M-15}$ and $bla_{NDM}$, and $P = 0.041$ for $bla_{OXA-48}$-like) (Table 4). GNB HSS colonisation appeared to be highest during the March–June and July–October periods.

### GNB species carrying $bla_{NDM}$ and $bla_{OXA-48}$-like

In total, 175 bacterial isolates from 151/6,290 HSS with a positive multiplex-PCR for a carbapenemase ARG were purified. MALDI-TOF MS identified 27 different bacterial species, with Klebsiella spp. and Enterobacter spp. carrying $bla_{NDM}$ and/or $bla_{OXA-48}$-like equally dominant ($n = 53/175$, 30.3%). Within the Klebsiella spp., K. pneumoniae was most prevalent, accounting for 86.8% ($n = 46/53$). Similarly, within the Enterobacter spp. isolates recovered, E. hormaechei was the most frequent ($n = 38/53$, 71.7%), followed by E. cloacae ($n = 8/53$, 15.1%). Other GNBs identified include Acinetobacter baumannii ($n = 13/175$, 7.4%), Pseudomonas spp. ($n = 14/175$, 8%), Serratia marcescens ($n = 12/175$, 6.9%) and Leclercia adecarboxylata ($n = 11/175$, 6.3%) (Fig. 2). Within the dataset, only three E. coli isolates carrying carbapenemase ARG were recovered ($n = 3/175$, 1.7%, Supplementary Data 4 and Supplementary Table 3).

$bla_{NDM}$-producing K. pneumoniae and E. hormaechei isolates were mostly found in Pakistan and altogether represented over 60% of the 93 carbapenemase-producing isolates found in PP (Supplementary Data 4). Supplementary Table 3 shows detailed information on surfaces where these isolates were recovered from.

### Bacterial diversity and whole genome sequencing (WGS) analysis of ARG variants

Whole genome sequencing (WGS) data was available for 128 carbapenemase-positive GNB isolates out of the 175 total isolates recovered from HSS (due to loss of growth or loss of carbapenemase gene upon regrowth), and of these, 18 bacterial species were identified (Supplementary Data 5).

Of the GNB isolates with WGS data, 122 carried at least one $bla_{NDM}$ variant ($n = 114$ $bla_{NDM-1}$, $n = 3$ $bla_{NDM-5}$, and $n = 7$ $bla_{NDM-7}$ genes) and 17 isolates carried $bla_{OXA-48}$-like genes ($n = 14$ $bla_{OXA-181}$, $n = 2$ $bla_{OXA-204}$, $n = 1$ $bla_{OXA-48}$). The majority of carbapenemase ARGs were found to be plasmid-mediated (Fig. 3), however, for 16 isolates, the ARGs were chromosomally located (Supplementary Data 6).

Twelve isolates concomitantly carried variants of $bla_{NDM}$ and $bla_{OXA-48}$-like. This was most notably detected within sequence type (ST) 15 K. pneumoniae ($n = 10/16$ from Pakistan), with $n = 9/10$ carrying both $bla_{NDM-1}$ and $bla_{OXA-181}$ on similar ~140 kbp IncA/C and ~50 kpb ColKP3-IncX3 hybrid plasmids respectively (Fig. 3, Supplementary Figs. 1–3); $n = 1/10$ carried both variants, but the plasmid type remained unknown due to assembly fragmentation.

Two ST11 K. pneumoniae isolates from Pakistan carried two $bla_{NDM}$ genes: one harboured two copies of $bla_{NDM-7}$ on a 30 kbp IncX3 and 145 kbp IncA/C plasmids, and the other one carried $bla_{NDM-1}$ (on a ~39 kpb IncX3 plasmid) and $bla_{NDM-7}$ on a ~145 kpb IncA/C plasmid (Fig. 3).

The other two Klebsiella species that were identified were K. quasipneumoniae and K. michiganensis. Plasmids represented in Supplementary Figs. 1, 2 from were from multiple bacterial species, including K. pneumoniae, K. michiganensis and Enterobacter roggenkampii.

Of the six isolates with $bla_{NDM-7}$, three were K. pneumoniae and three were E. hormaechei.

From the 47 Enterobacter isolates, E. hormaechei was dominant ($n = 38$). Supplementary Data 6 shows the distinct ST recovered, with

**Table 1 | Gram-negative bacteria (GNB) colonisation per hospital sites and countries**

| Country/hospital | Samples with GNB colonisation | No growth | Total |
|---|---|---|---|
| Bangladesh | 944 (92.19%) | 80 (7.81%) | 1024 |
| BC | 371 (90.49%) | 39 (9.51%) | 410 |
| BK | 573 (93.32%) | 41 (6.68%) | 614 |
| Ethiopia (ES) | 119 (47.04%) | 134 (52.96%) | 253 |
| Nigeria | 927 (59.2%) | 639 (40.8%) | 1566 |
| NK | 435 (92.16%) | 37 (7.84%) | 472 |
| NN | 273 (39.28%) | 422 (60.72%) | 695 |
| NW | 219 (54.89%) | 180 (45.11%) | 399 |
| Pakistan (PP) | 642 (62.15%) | 391 (37.85%) | 1033 |
| Rwanda | 538 (58.04%) | 389 (41.96%) | 927 |
| RK | 317 (67.88%) | 150 (32.12%) | 467 |
| RU | 221 (48.04%) | 239 (51.96%) | 460 |
| South Africa (ZAT) | 646 (43.44%) | 841 (56.56%) | 1487 |
| Total | 3816 (60.67%) | 2474 (39.33%) | 6290 |
| Significance | $P = 1.329^{-134}$ | | |

Total number of samples collected per country and per hospital site were used as denominators to calculate the percentages. Additionally, a Chi-squared test was conducted to assess the extent of the overall differences in the proportions of GNB colonisation between the countries. GNB colonisation varied significantly ($P < 0.001$) between the six countries. $P$ value of <0.05 was considered to show significant differences between the proportions. Abbreviations for BARNARDS hospitals detailed in the "Methods" section.

21 STs for *Enterobacter* alone, with the most frequent being ST120, ST231, ST316 and ST418, all *E. hormaechei*. Three *E. hormaechei* isolates with different STs carrying a $bla_{NDM-5}$ on IncX3 plasmids (ranging from 46 to 55 kb; Fig. 3) were recovered from a single hospital (NK) in the same month in 2016 (Supplementary Data 5 shows the recovery date of the isolates with WGS data available). The two 46 kb plasmids (pNK-E166-IncX3 and pNK-E171A-IncX) shared 99.7% sequence homology. Supplementary Figs. 3, 4 show also that the 55 kb pNK-E179A-IncX3 was more genetically distant (30–31% aligned homologous sequences). Eight *Enterobacter* spp. isolates, mostly ST316 *E. hormaechei* ($n = 5$) co-carried $bla_{NDM-1}$ and *mcr-9* on a large 400kbp IncHI2 plasmid. *S. marcescens* was most often recovered in both BC and BK, all $n = 9$ isolates carried $bla_{NDM-1}$, either on IncF or IncL/M plasmid types (Fig. 3).

One ST78 *A. baumannii* with two copies of $bla_{NDM-1}$ and four ST52 *A. baumannii* isolates were recovered from RU and the hospital in South Africa (ZAT). Of the two *E. coli* isolates (ST448 and ST405) identified by WGS, both carried $bla_{NDM-1}$ and one (ST405) co-carried $bla_{NDM-1}$ and $bla_{OXA-181}$.

WGS confirmed the presence of multiple aminoglycosides (*aac, ant, aph*), ESBLs ($bla_{OXA-1}$, $bla_{SHV-11}$, $bla_{SHV-12}$, $bla_{SHV-106}$, $bla_{SHV-182}$ and $bla_{SHV-187}$), fosfomycin (*fosA*) and tetracycline (*tetA, tetB, tetD*) ARGs. All isolates were additionally screened for genes conferring resistance to disinfectants, and *MexAB-OprM, MexCD-OprJ, MexEF-OprN* and *MexJK-OpmH* were identified in three *Pseudomonas* spp. isolates, two from Bangladesh and one from Pakistan.

### Evidence of local transmission and links to neonatal sepsis

From WGS and epidemiology analysis, nine potential cluster/transmission events were analysed by SNP analysis (Fig. 4). HSS isolates from two clusters shared the same ST of isolates from the same hospital site causing neonatal sepsis during BARNARDS (ST15 *K. pneumoniae* and ST52 *A. baumannii* isolates). WGS data from these sepsis isolates were included in the single nucleotide polymorphism analysis (total of $n = 22$ ST 15 *K. pneumoniae* isolates causing sepsis in Pakistan)[26].

For ST15 *K. pneumoniae*, there were 16 HSS isolates analysed alongside 22 ST15 *K. pneumoniae* isolates from neonatal blood cultures within PP during the same time-period (November 2015–November 2017). ST15 *K. pneumoniae* were recovered from HSS from emergency neonatal care ($n = 5$), patient's zone ($n = 4$), ward furniture/surfaces ($n = 1$), medical equipment ($n = 1$), and NA ($n = 5$). SNP analysis revealed that, except for one isolate, ST15 *K. pneumoniae* from Pakistan from both HSS and neonatal sepsis blood cultures ($n = 22$) were within 10 pairwise SNPs. Interestingly, in eight HSS isolates, $bla_{OXA-181}$ and the ColKP3–IncX3 hybrid replicon were not detected, indicating that some isolates might not have acquired the ColKP3-IncX3 plasmid or may have lost the plasmid during culture for WGS. Furthermore, the ST405 *E. coli* isolates from the same hospital wards in Pakistan carried genetically similar plasmids, both the IncA/C $bla_{NDM-1}$ and the ColKP3-IncX3 $bla_{OXA-181}$ plasmids suggesting plasmid transmission may have occurred between bacterial species colonising hospital surfaces. It is possible that the ST405 *E. coli* isolate acquired both $bla_{NDM-1}$ and $bla_{OXA-181}$ from ST15 *K. pneumoniae*, as our data indicates multiple hospital surfaces were contaminated with this strain between 2016 and 2017 (Supplementary Figs. 5–8).

From nine *A. baumannii* isolates with WGS data, five were ST52, and SNP analysis of ST52 revealed the sepsis isolate (identified in the same hospital in Rwanda (RU)[26]) to be genetically distant (>100 SNPs) from the isolates collected from HSS, which were within 10 SNPs.

Isolates of ST20 and ST1317 $bla_{NDM-1}$ *K. pneumoniae* were within 10 pairwise SNPs among isolates from each cluster, further evidencing the colonisation and potential transmission across the hospital wards in Pakistan. For ST20 isolates, there were two small sub-clusters identified, with genetic grouping occurring for isolates collected in late 2016 (September–November) and a distinct cluster of over 1000 SNPs distinct for ST20 *K. pneumoniae* recovered between January and March 2017 (Fig. 4).

*E. hormaechei*, the dominant *Enterobacter* species, was mostly identified in Asia. Clusters of ST231 and ST316 ($bla_{NDM-1}$ and *mcr-9*) *E. hormaechei* isolates recovered across surfaces in PP were within three and two pairwise SNPs. A cluster of seven ST418 *E. hormaechei* HSS isolates from BK, recovered during a 6-week period between October and November 2017, were within four pairwise SNPs.

A cluster of five BK *S. marcescens* isolates (within 3 SNPs) was over 10,000 SNPs distant from the *S. marcescens* isolates from NN and BC. Nine *L. adecarboxylata* isolates isolated between March and November 2017 from different surfaces in PP were closely related (within a single SNP distance) (Fig. 4).

## Discussion

Herein, we report a high prevalence of bacteria carrying $bla_{CTX-M-15}$, $bla_{NDM}$ and $bla_{OXA-48}$-like genes colonising environmental surfaces and patient care equipment in 10 hospitals across six LMICs. To the best of our knowledge, this multinational study is the first to evidence transmission networks between hospital wards and neonates with sepsis in LMICs[26]. Importantly, a high-risk double carbapenemase ST15 *K. pneumoniae* clone[8,27] was recovered from HSS and blood cultures from septic neonates in Pakistan over a 2-year period, suggesting this strain was colonising surfaces in the ward whilst simultaneously causing neonatal sepsis[26]. In our dataset, we identified a single *E. coli* isolate carrying a genetically similar $bla_{NDM}$ (IncAC plasmid type) plasmid to the ST15 *K. pneumoniae* strain isolated from the same hospital (PP). We evidence similarity between plasmids detected within *Enterobacterales* from multiple hospital surfaces sampled between May 2016 and November 2017, indicating possible horizontal transmission however, a larger genomics dataset would be needed to understand plasmid transmission dynamics upon hospital surface samples (e.g. plasmids harboured in *E. hormaechei* isolates from NK).

ST15 *K. pneumoniae* has been reported to be responsible for neonatal sepsis and high mortality in clinical settings[8], is often linked

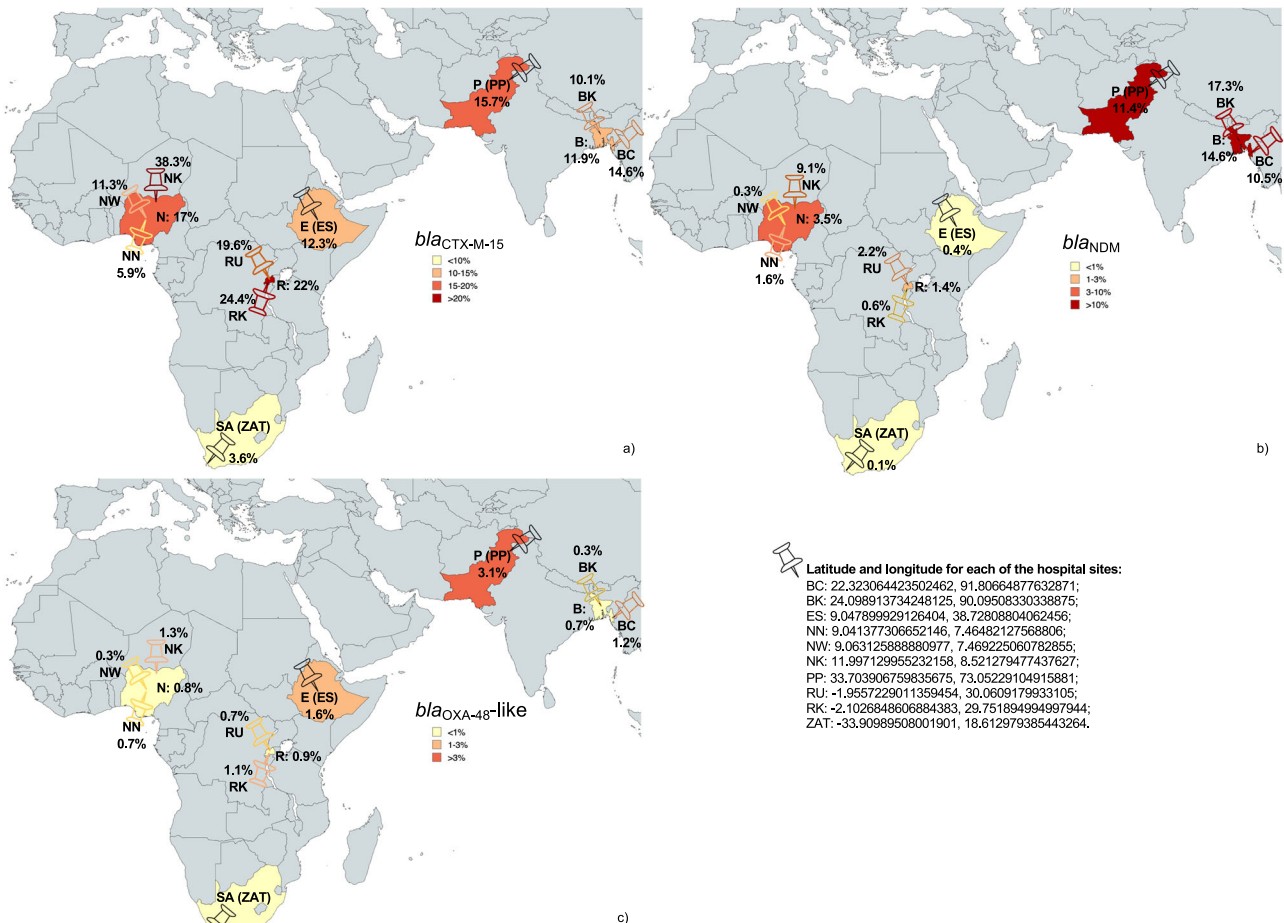

**Fig. 1 | Maps indicating prevalence of ARG per site and country.** Prevalence of $bla_{\text{CTX-M-15}}$ (**a**), $bla_{\text{NDM}}$ (**b**), and $bla_{\text{OXA-48}}$-like (**c**) genes per country and per hospital site. The total number of samples collected per country and per hospital site were used as denominators to calculate the prevalence of each antibiotic resistance gene (ARG) per country and per hospital site, respectively. Abbreviations for BARNARDS hospitals are detailed in the "Methods" section, map pins showing latitude and longitude, and coloured according to the ARG prevalence in that site. Source data are provided in Supplementary Table 2.

with HAIs and hospital outbreaks[28–30] and is frequently detected in the environment (wastewater and soil) in Africa and Asia[31,32] Furthermore, multiple clusters of the same strain of different GNB species were detected from hospital surfaces in Bangladesh and Rwanda, suggesting that hospital surface colonisation by pathogenic bacteria is a significant concern in LMICs hospitals. WHO and Médecins Sans Frontières still lack AMR data from many LMIC settings, thus, global reports on IPC, burden of sepsis or HAIs contain information mostly generated from HICs. This data highlights the widespread bacterial colonisation and the transmission of bacteria carrying multiple ARG upon hospital surfaces, which could be useful to guide realistic approaches and support action plans for countries where IPC practices are limited[13,18,19,22,33,34].

Previous studies report that frequently used medical equipment/ touch surfaces in healthcare settings are crucial for the cross-transmission of AMR bacteria and HAIs, such as sepsis[3,7,17,21,22,25,35]. This is particularly true in institutions where resources to implement IPC programs are scarce[4,13]. Bacteria carrying ARGs appeared to be found most frequently on surfaces near the sink drain, which was concordant with the work by Firesbhat et al. in Ethiopia[16]. We also found contamination of medical equipment and/or ward furniture/ surfaces, as detected in other African-based studies reporting on HAIs[24,36–38].

Interestingly, ESBL- and carbapenemase-producing *Enterobacterales* were more frequently detected in HSS collected between March and October. Despite the potential contamination pattern during this period, due to inconsistent sampling throughout the hospital sites among time periods, detailed seasonal analysis per country could not be performed herein. Apart from unequal sample size, a variety of factors, including temperature, cleaning practices, or healthcare workers shifts, may be influencing bacterial reservoirs at particular times of the year.

Our findings are comparable with previous studies in Pakistan[25], Nigeria[36] and Ethiopia[24,39], which report >60% bacterial growth on medical equipment and environmental surfaces. We observed large differences in GNB growth between countries or hospitals within the same country, emphasising that bacterial colonisation should be monitored in each hospital. Apart from the sampling limitations of this study, there are likely various factors contributing to the range of colonisation observed, including antibiotic accessibility[40].

In this study, *K. pneumoniae*, *E. hormaechei*, *A. baumannii*, *S. marcescens* and *L. adecarboxylata* were the most prevalent bacterial species carrying $bla_{\text{CTX-M-15}}$, $bla_{\text{NDM}}$ and $bla_{\text{OXA-48}}$-like genes across all countries, which have been commonly described[1,3,8,11,17,24,25,35,39]. WGS revealed diversity within species showing multiple and co-occurring dominant STs of *E. hormaechei* and *K. pneumoniae*. MDR *A. baumannii* carrying OXA-carbapenemase genes were found among hospital surfaces in Bangladesh in 2021[23]. In this study, however, *E. hormaechei*, *Pseudomonas* spp. and *Shewanella putrefaciens* were identified as the bacterial species carrying $bla_{\text{OXA-48}}$-like genes in BK. Whilst we did

**Table 2 | Number of hospital surface swabs (HSS) collected per country and per hospital site, over the total of 4126 HSS (where information regarding surface was available)**

| Country | Hospital site | Patient's zone | Surfaces near the sink drain | Emergency neonatal care | Ward furniture/ surfaces | Mobile medical equipment | Medical equipment | Total |
|---|---|---|---|---|---|---|---|---|
| Bangladesh | BC | 1 | 73 | 11 | 73 | 78 | 19 | 255 |
| | BK | 82 | 26 | 0 | 78 | 57 | 1 | 244 |
| | Total B | 83 | 99 | 11 | 151 | 135 | 20 | 499 |
| Ethiopia | ES | 41 | 54 | 32 | 80 | 16 | 22 | 245 |
| | Total E | 41 | 54 | 32 | 80 | 16 | 22 | 245 |
| Nigeria | NK | 28 | 5 | 15 | 19 | 11 | 34 | 112 |
| | NN | 139 | 67 | 133 | 83 | 90 | 146 | 658 |
| | NW | 202 | 12 | 89 | 47 | 10 | 16 | 376 |
| | Total N | 369 | 84 | 237 | 149 | 111 | 196 | 1146 |
| Pakistan | PP | 160 | 42 | 121 | 76 | 135 | 247 | 781 |
| | Total P | 160 | 42 | 121 | 76 | 135 | 247 | 781 |
| Rwanda | RK | 50 | 38 | 179 | 69 | 50 | 15 | 401 |
| | RU | 35 | 37 | 131 | 79 | 58 | 54 | 394 |
| | Total R | 85 | 75 | 310 | 148 | 108 | 69 | 795 |
| South Africa | ZAT | 87 | 105 | 22 | 152 | 42 | 252 | 660 |
| | Total SA | 87 | 105 | 22 | 152 | 42 | 252 | 660 |
| TOTAL | | 825 | 459 | 733 | 756 | 547 | 806 | 4,126 |

Abbreviations for BARNARDS hospitals are detailed in the "Methods" section.

**Table 3 | Statistical analysis to identify correlation between hospital surfaces and Gram-negative bacteria (GNB) colonisation and antibiotic resistance genes (ARGs)**

| (a) | GNB colonisation | | |
|---|---|---|---|
| | Positive | Negative | Total |
| Patient's zone | 511 (61.94%) | 314 (38.06%) | 825 |
| Surfaces near the sink drain | 314 (68.41%) | 145 (31.59%) | 459 |
| Emergency neonatal care | 379 (51.71%) | 354 (48.29%) | 733 |
| Ward furniture/surfaces | 474 (62.7%) | 282 (37.3%) | 756 |
| Mobile medical equipment | 343 (62.71%) | 204 (37.29%) | 547 |
| Medical equipment | 370 (45.91%) | 436 (54.09%) | 806 |
| Total | 2391 (57.95%) | 1735 (42.05%) | 4126 |

| (b) | | $bla_{CTX-M-15}$ | | $bla_{NDM}$ | | $bla_{OXA-48}$-like | | |
|---|---|---|---|---|---|---|---|---|
| | | Negative | Positive | Negative | Positive | Negative | Positive | Total |
| Patient's zone | Count | 702 | 123 | 786 | 39 | 815 | 10 | 825 |
| | Row N % | 85.1% | 14.9% | 95.3% | 4.7% | 98.8% | 1.2% | 100.0% |
| Surfaces near the sink drain | Count | 365 | 94 | 425 | 34 | 444 | 15 | 459 |
| | Row N % | 79.5% | 20.5% | 92.6% | 7.4% | 96.7% | 3.3% | 100.0% |
| Emergency neonatal care | Count | 607 | 126 | 705 | 28 | 724 | 9 | 733 |
| | Row N % | 82.8% | 17.2% | 96.2% | 3.8% | 98.8% | 1.2% | 100.0% |
| Ward furniture /surfaces | Count | 652 | 104 | 725 | 31 | 750 | 6 | 756 |
| | Row N % | 86.2% | 13.8% | 95.9% | 4.1% | 99.2% | 0.8% | 100.0% |
| Mobile medical equipment | Count | 476 | 71 | 521 | 26 | 543 | 4 | 547 |
| | Row N % | 87.0% | 13.0% | 95.2% | 4.8% | 99.3% | 0.7% | 100.0% |
| Medical equipment | Count | 723 | 83 | 773 | 33 | 797 | 9 | 806 |
| | Row N % | 89.7% | 10.3% | 95.9% | 4.1% | 98.9% | 1.1% | 100.0% |
| **Total** | Count | 3525 | 601 | 3935 | 191 | 4073 | 53 | 4126 |
| | Row N % | 85.4% | 14.6% | 95.4% | 4.6% | 98.7% | 1.3% | 100.0% |
| Significance | | $P = 0.000013$ | | $P = 0.071$ | | $P = 0.004$ | | |

Chi-squared test was conducted to assess the extent of the differences in the proportions of ARGs between the surface categories. (a) Frequency of positive and negative GNB growth among hospital surfaces. (b) Contingency table showing the percentage of negative and positive samples for ARG among hospital surfaces. Significant differences ($P < 0.05$) were observed when comparing the prevalence of each ARG across surfaces; $bla_{CTX-M-15}$ and $bla_{OXA-48}$-like frequency varied significantly between the six surface categories ($P < 0.05$), but no significant differences in presence across surfaces were observed for $bla_{NDM}$ ($P > 0.05$).

**Table 4 | Distribution of antibiotic resistance genes (ARGs) prevalence and Gram-negative bacteria (GNB) colonisation according to the timeline (November 2015–January 2018)**

| Timeline | $bla_{\text{CTX-M-15}}$ | $bla_{\text{NDM}}$ | $bla_{\text{OXA-48}}$-like | GNB colonisation | Total HSS collected in each time band |
|---|---|---|---|---|---|
| Nov15–Feb16 | 32 (10.36%) | 16 (5.18%) | 2 (0.65%) | 159 (51.46%) | 309 |
| Mar16–Jun16 | 110 (15.87%) | 42 (6.06%) | 12 (1.73%) | 456 (65.8%) | 693 |
| Jul16–Oct16 | 94 (11.49%) | 30 (3.67%) | 6 (0.73%) | 459 (56.11%) | 818 |
| Nov16–Feb17 | 61 (6.02%) | 57 (5.62%) | 10 (0.99%) | 534 (52.66%) | 1014 |
| Mar17–Jun17 | 126 (18.31%) | 84 (12.21%) | 16 (2.33%) | 501 (72.82%) | 688 |
| Jul17–Oct17 | 188 (21.05%) | 43 (4.82%) | 7 (0.78%) | 623 (69.76%) | 893 |
| Nov17–Jan18 | 28 (11.34%) | 16 (6.48%) | 4 (1.62%) | 141 (57.09%) | 247 |

Chi-squared test was conducted to assess the extent of the differences in the proportions of ARGs between the time bands. The total of samples collected during each time band were used as denominators to calculate the percentages. The percentage of the ARGs varied significantly between the time bands ($P = 2.89^{-22}$ for $bla_{\text{CTX-M-15}}$, $P = 2.82^{-10}$ for $bla_{\text{NDM}}$, and $P = 0.041$ for $bla_{\text{OXA-48}}$-like). GNB colonisation also varied significantly ($P = 4.98^{-25}$). P value was considered statistically significant when $P < 0.05$.

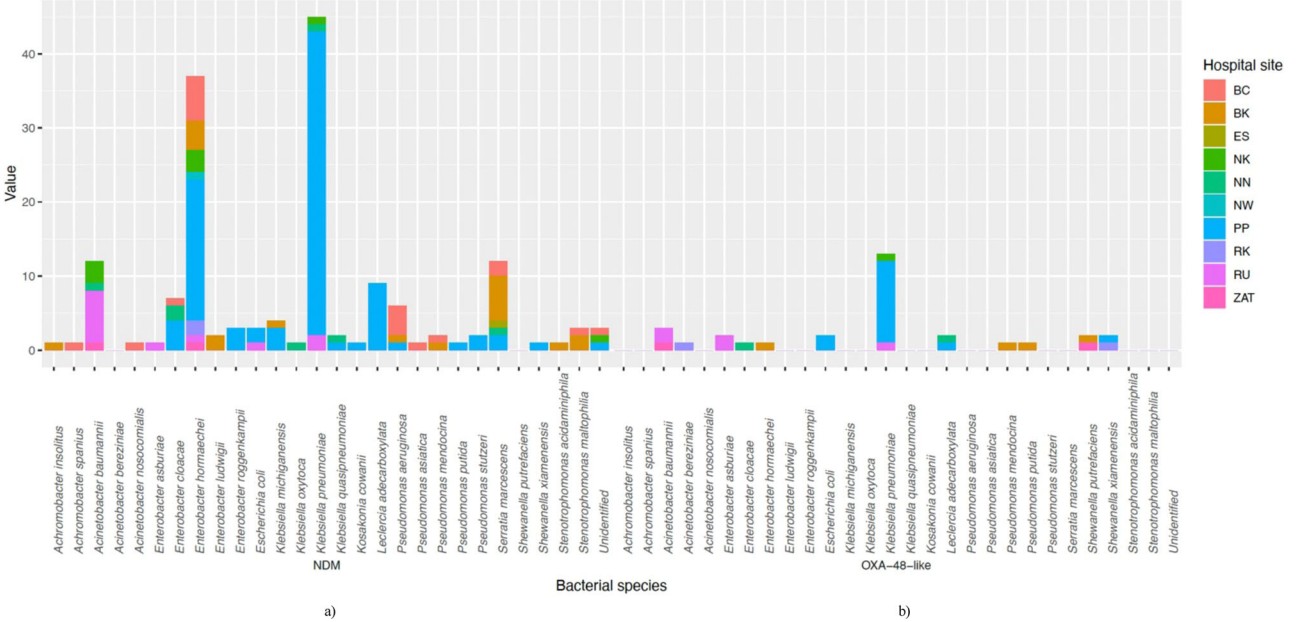

**Fig. 2 | Bacterial species diversity carrying carbapenemase genes.** Bacterial species carrying $bla_{\text{NDM}}$ (**a**) and $bla_{\text{OXA-48}}$-like (**b**) among countries and sites, according to PCR screening and MALDI-TOF MS identification (a total of 175 bacterial isolates from 27 bacterial species; and 3 "unidentified" isolates). Source data are provided in Supplementary Data 4.

detect *A. baumannii* (7.4%), this was mostly recovered from HSS collected in RU (ST52 carrying $bla_{\text{NDM-1}}$), but none from Bangladesh. The present work confirms that $bla_{\text{OXA-48}}$-like genes are widespread in African and South-Asian countries. Contrary to results from a hospital surface screen of multiple wards in a hospital in Ghana[1], we showed a lower prevalence of $bla_{\text{NDM}}$ ($n = 338/6290$ samples, 5.4%). Nevertheless, the prevalence of this gene, particularly in Pakistan ($n = 162/1033$, 15.7%), was in line with a study from several hospitals in the same country[41]. $bla_{\text{KPC}}$ is not common in Africa or South Asia[7,26], which this study also confirmed; however, $bla_{\text{KPC}}$ was detected from neonatal cots in Tanzania in 2021[17] and $bla_{\text{KPC}}$ *K. pneumoniae* was found on pillows and the floor in three hospitals in Bangladesh[42].

Limitations of this study include a lack of consistency in swabbing areas, as their descriptions were, on occasions, imprecise and the samples varied across each hospital. Thus, we were cautious with data interpretation and did not present correlation data between GNB growth and/or ARG prevalence per surface per hospital or between surface type/category and bacterial species due to sampling limitations. A second limitation was not grouping swab locations to reduce

the testing size, which resulted in a total of 309 different surfaces, causing difficulties when comparing across countries and hospitals. Thirdly, there were differences in the number of samples collected across hospital sites and time, as we did not confirm the exact number/type of samples per month. Moreover, there was a lack of written information accompanying some samples—not all the samples were well-described. Information related to cleaning and disinfection practices in each hospital site was not collected, which might explain some of the results obtained; PCR screening for other genes would have also given us a broader view of potential resistance to other antibiotics available in LMICs but not commonly used due to economic resources or accessibility.

To summarise, we have shown that ESBL- and carbapenemase-producing *Enterobacterales* were most prevalent in samples collected near the sink drain. Transmission events occurred across patient care equipment and environmental surfaces of the hospital wards, and worryingly, there was evidence that the same strains have caused neonatal sepsis. Moreover, and of particular concern, is the high prevalence of antibiotic resistance determinants and the diversity of ARG-

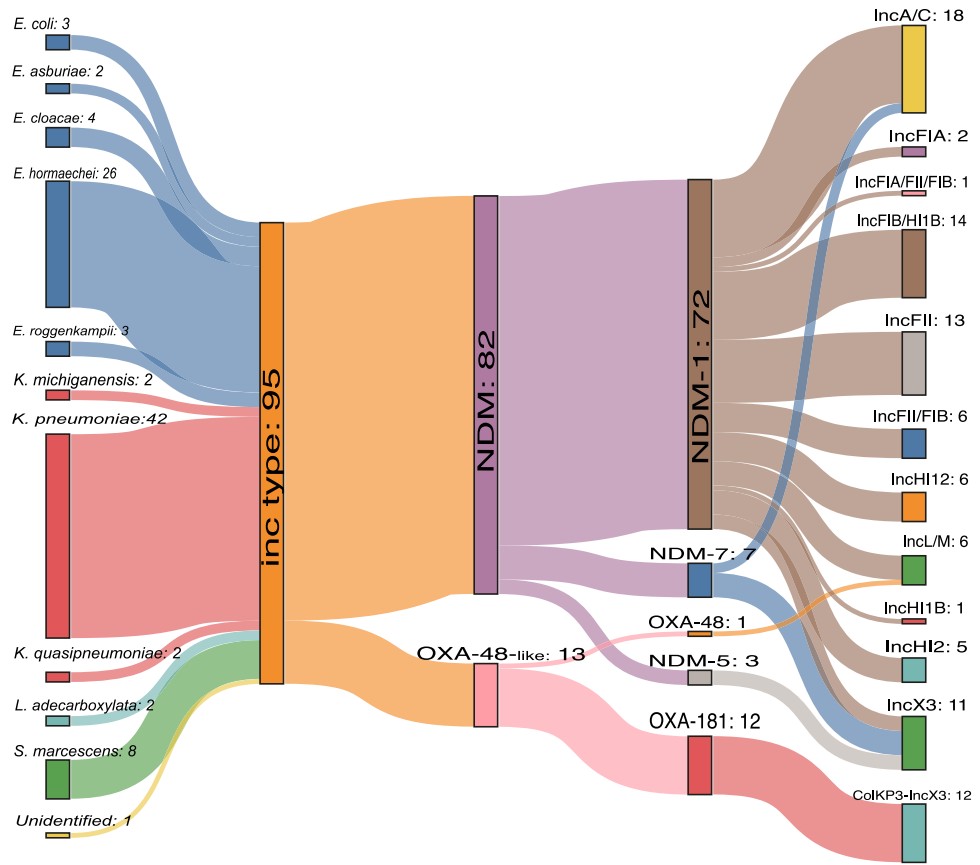

**Fig. 3 | A Sankey diagram linking bacterial species, plasmids and carbapenemase ARGs.** There were *n* = 95 isolates with an identifiable Inc plasmid type (due to assembly fragmentation, it was not always possible to assemble and analyse whole plasmids). The carbapenemase variant NDM and OXA-48-like group are divided into variants, and the Inc type detected per ARG-variant is shown. Source data are provided in Supplementary Data 5 and 6 and as Source Data file.

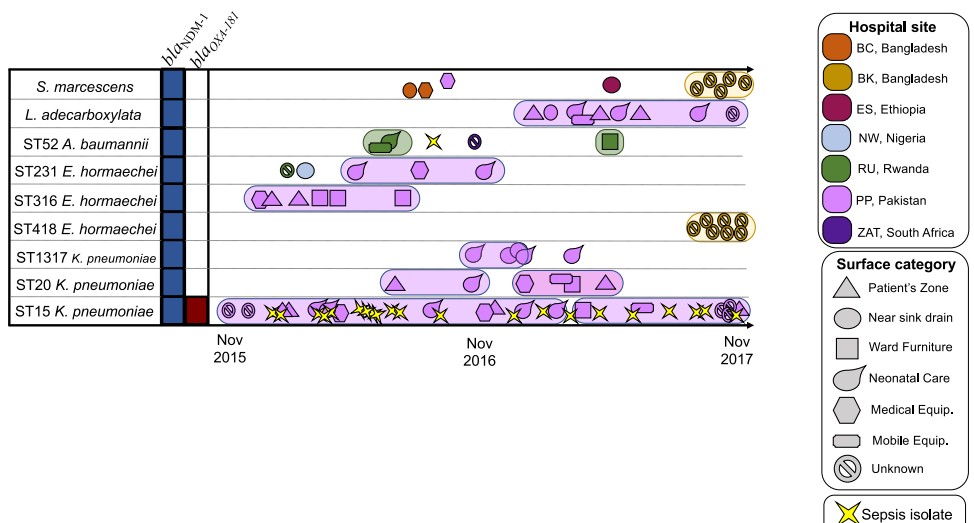

**Fig. 4 | Timeline showing nine potential clusters of bacterial isolates recovered from hospital surface swabs (HSS) and whether the same strain of bacteria was found in sepsis isolates during BARNARDS.** Sequence types (STs) and antimicrobial resistance genes (ARGs) are shown, and the highlighted background around isolates indicated these were within 10 pairwise SNPs. The different pink tone for ST20 *K. pneumoniae* indicates distinct sub-clusters identified among these isolates in Pakistan (over 1000 SNPs). Abbreviations for BARNARDS hospitals are detailed in the "Methods" section. The figure was created using Adobe Illustrator v26.5. Source data are provided in Supplementary Data 5 and 6.

containing bacteria among surfaces within the hospital facilities, presenting an increasing threat to the patients. Future work to determine the relative risk of inborn neonates developing sepsis with specific bacterial strains that are colonising HSS is warranted to fully understand the impact of bacterial transmission events across the wards in neonatal sepsis. Our results further emphasise the extent of hospital surface contamination with bacteria carrying multiple ARG in LMICs, calling for an urgent assessment of improved IPC practice compliance and tailored guidelines for each hospital site.

## Methods

### Settings, ethics, participants, and study design

The BARNARDS network included Bangladesh, Chittagong Maa-O-Shishu Hospital, Chattogram (BC), and Kumudini Women's Medical College, Mirzapur (BK); Ethiopia, St. Paul's Hospital Millennium Medical College, Addis Ababa (ES); India, Division of Bacteriology, ICMR-National Institute of Cholera and Enteric Diseases Beliaghata and Institute of Post-Graduate and Medical Education and Research, Kolkata (IN); Nigeria, National Hospital Abuja (NN), Wuse District Hospital, Abuja (NW), and Murtala Mohammad Specialist Hospital, Kano (NK); Pakistan, Pakistan Institute of Medical Sciences (PP) and Bhara Kahu Rural Health Centre, Bhara Kahu (PC); Rwanda, University Central Hospital of Kigali, Kigali (RU) and Kabgayi Hospital, Kabgayi (RK); South Africa, Tygerberg Hospital, Cape Town (ZAT). HSS were not collected in India or one of the hospitals in Pakistan (PC). The hospital site abbreviation names are used throughout this article; however, the country name is used when the results are applicable to all hospitals within that country.

Standard operating procedures (SOPs) were designed and adhered to throughout the network (https://www.ineosoxford.ox.ac.uk/research/areas-of-focus/amr-burden/barnards), and ethical approval was obtained from local ethics committees prior to the start of the study.

From November 2015 until January 2018, HSS were collected from different surfaces (Supplementary Data 1) within different wards (i.e. maternity and neonatal intensive care units). Frequently, information regarding wards was incomplete, thus, only the surface but not their location within the hospitals was considered for analysis. Samples from environmental surfaces and patient care equipment in hospital settings were collected with charcoal swabs and stored at 4 °C until transported to the UK under UN3373 conditions.

### HSS processing

HSS were streaked on three chromogenic agar media plates (Liofilchem®, Italy) supplemented with vancomycin (10 mg/L) to promote the growth of GNB, vancomycin and cefotaxime (VC, 10 and 1 mg/L, respectively) to select for ESBL producers, and vancomycin and ertapenem (VE, 10 and 2 mg/L, respectively) to select for carbapenemase producers[7]. VC plates were tested for the presence of $bla_{CTX-M-15}$ by PCR and VE plates were tested for $bla_{NDM}$, $bla_{OXA-48}$-like and $bla_{KPC}$ by multiplex-PCR. All bacterial cultures were preserved in TS/72 cryogenic beads (Technical Service Consultants, UK) and stored at −80 °C. When VE bacterial growth yielded multiplex-PCR positive results, phenotypically distinct bacterial colonies were isolated and screened for the carbapenemase genes in the study by multiplex-PCR. Those isolates with a positive result were identified by MALDI-TOF MS (Bruker Daltonik GmbH, Coventry, UK) and preserved as detailed above. As per the BARNARDS protocol, due to the high prevalence of $bla_{CTX-M-15}$, we did not scrutinise samples for $bla_{CTX-M-15}$ positive isolates. The study workflow and dataset are summarised in Fig. 5. ARG prevalence resulting from PCR screening was represented in Coloured maps, which were created using MapChart (https://www.mapchart.net).

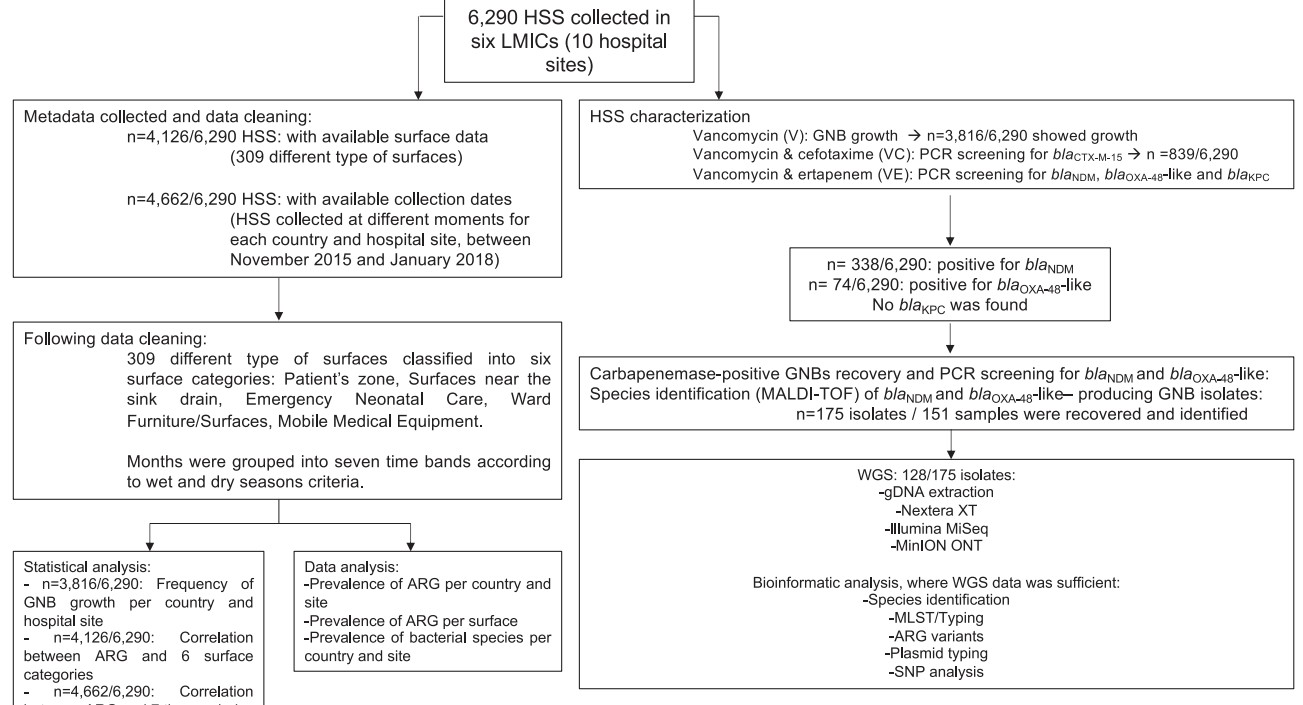

**Fig. 5 | Workflow of sample collection and processing.** Diagram detailing the total number of hospital surface swabs (HSS) collected, showing Gram-negative bacteria (GNB) growth, and screened for the presence of $bla_{CTX-M-15}$, $bla_{NDM}$, $bla_{KPC}$ and $bla_{OXA-48-like}$ antimicrobial resistance genes (ARGs), the number of GNB isolates recovered carrying carbapenemase genes, the number of isolates characterised by whole genome sequencing (WGS) and, where WGS data was sufficient, bioinformatic analysis was performed. Isolates for WGS were chosen after culture on VE (vancomycin, ertapenem) agar. Recoverable isolates after −80 °C preservation were selected for gDNA extraction and WGS. Data cleaning is also detailed; where data regarding hospital surfaces and collection dates was available, statistical analysis and data analysis were performed.

## Whole genome sequencing (WGS)

WGS was carried out on GNB isolates positive for at least one of the carbapenemase genes. Briefly, and as detailed by Sands et al.[26], gDNA was extracted using the QIAmp DNA mini kit (Qiagen, Germany) on the QIAcube platform (Qiagen, Germany) and quantified using the Qubit fluorometer 4.0. Short-read sequencing libraries were prepared using the Nextera XT v2 kit and paired-end sequenced on an Illumina MiSeq using the V3 chemistry to generate fragment lengths up to 300 bp (600 cycles). Long reads were prepared using Oxford Nanopore Technology (ONT) and libraries were generated using the 96-Rapid Barcoding Kit (SQK-RBK110.96; ONT). Sequencing was performed using MinION flow cells (R9.4.1) for a running time of 72 h within MinKnow.

## Bioinformatic analysis

To determine whether bacterial transmission events occurred within the hospital wards and/or neonatal sepsis, bioinformatic analysis was performed as detailed by Sands K et al.[26]. ONT FAST5 reads were base called using Guppy v5.0.11 and NVIDIA V100 GPUs. Following QC on short reads (Illumina) and long reads (ONT) as described by Boostrom et al.[43], reads were assembled using Unicycler (v0.4.9). Genome assembly metrics were generated using QUAST (v.5.2.0)[44] and bacterial species were identified using Pathogenwatch[45]. Multilocus sequence typing (MLST), characterisation of genes conferring resistance to antibiotics and/or disinfectants genes and plasmid genomic profiles were performed using ABRicate (v0.9.7)[46] and associated databases: NCBI[47], PlasmidFinder[48] and CARD[49]. Previously undefined alleles and sequence type (ST) profiles were submitted to Enterobase, BIGSbd and PubMLST for assignment[50]. Genomes were annotated using Prokka (v1.14.5)[51]. Following initial WGS analysis, four or more isolates with the matching ST/species from a single hospital site were considered a potential cluster and were subject to SNP analysis using snippy (v4.6.0)[52] with BWA and freebayes mapping the reads and calling variants --min cov 20. To maximise SNP calling, a high-quality internal reference was used[53]. A pairwise SNP matrix was performed converting a FASTA alignment to an SNP distance matrix using Snp-dists (v0.8.2)[54]. Adobe Illustrator v26.5 was used to generate the isolate-relatedness timeline figure. The Sankey diagram was created using Sankeymatic.com.

For plasmid analysis for evidence of transmission between bacterial isolates/species, the contig containing the carbapenemase variant and plasmid was analysed using the Mauve aligner[55] within Geneious (v2023.2.1). The size of the plasmid and whether the assembler theoretically denoted the contig as circular and complete were recorded. Following Mauve alignment, the plasmid sequences were annotated using Prokka (v1.14.5) and further annotated using a reference plasmid from a BLAST search within Geneious. Visualisations of annotated aligned plasmids were performed within Geneious. Further similarity matrices were generated using mash dist (v2.2)[56] to estimate the genetic similarity between plasmid sequences. Sequence alignments were performed within Geneious using the MAFFT aligner plugin.

## Data cleaning and statistical analysis

To analyse the frequency of GNB growth in hospital surfaces, GNB growth on plates supplemented with vancomycin was examined, and the total number of processed samples was used as a denominator for the prevalence of assessed ARGs. HSS was collected from different surfaces (Supplementary Data 1) and included both environmental surfaces and patient care equipment in hospital settings, which were classified into six different categories (Supplementary Data 7) based on WHO and CDC IPC guidelines and previous publications[4,13,18,57], following data cleaning. In this study, environmental surfaces included the hands of healthcare workers, which were categorised into the patient's

zone as they can act as a source of transmission[4,58,59]. The category of patient care equipment was defined as medical equipment or mobile medical equipment, categories that include non-critical, semi-critical and noncritical patient care equipment. The six final categories were: surfaces from the patient's zone (immediate environment), surfaces near the sink drain (including sink basin, faucet, faucet handles, and surrounding countertop)[60,61], emergency neonatal care, ward furniture/surfaces, mobile medical equipment, and medical equipment. Due to unequal sample size (Table 2) as well as sample variety (Supplementary Data 1), statistical analyses were performed using the total number of samples included in each surface category and not per country/hospital. Furthermore, following data cleaning, the HSS collection dates were classified into seven-time bands (Supplementary Table 4) according to Climate Change Knowledge Portal (World Bank Group)[62]. Statistical analysis to study the ARG prevalence was performed considering HSS collected per time band as a denominator, inclusive of all countries and hospitals. No seasonal analysis per country was performed.

ARG frequencies per hospital, country and surface category and corresponding figures were assessed. RStudio ggplot2 package was used for figure creation (RStudio version 4.3.0 (2023-04-21) -- "Already Tomorrow"), and IBM SPSS Statistics (Version 25.0.0.1) (190) was used to perform the statistical analyses. Statistical analyses were performed to determine whether certain surfaces were at greater risk of colonisation with β-lactamase-producing bacteria. The relationship between ARG and a surface category, as well as the frequency of GNB colonisation of hospital surfaces among countries and hospital sites, and over the timeline, were analysed to obtain counts and percentages. Chi-Squared tests were conducted to test the independence of the variables on contingency tables to establish if the overall differences in frequencies between GNB colonisation and ARG prevalence over countries and surfaces were statistically significant at the $P < 0.05$ level. When such differences were seen, the individual proportions for each of the countries and surfaces were examined and compared to determine where the main differences lay.

## Reporting summary

Further information on research design is available in the Nature Portfolio Reporting Summary linked to this article.

## Data availability

Databases used for in silico analysis in this work are PlasmidFinder, CARD, Enterobase, and PubMLST. Coloured maps in the paper were created using MapChart (https://www.mapchart.net). The dataset generated in this study is deposited in the Figshare repository (https://doi.org/10.6084/m9.figshare.23790360). Source data are provided with this paper; available as Supplementary Data files or provided as a Source Data file. Genomes are available in the NCBI database under BioProject number PRJNA971772 (and accession codes/accessible links are provided in Supplementary Data 8). The plasmid analysis data generated in this study for evidence of transmission is available in Supplementary figures. Source data are provided with this paper.

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

## Acknowledgements

We thank all enroled participants and their families. This work was supported by a combination of two research awards (nos. OPP1119772 and OP1191522) from the Bill & Melinda Gates Foundation. We thank Liofilchem® for their continued support in the distribution of their microbiology products to enable standardisation of standard operating procedures across the clinical sites. We thank J. Parkhill for their advice and guidance regarding the phylogenetic analyses. We thank Wales Gene Park and ARCCA for their continued bioinformatics support and infrastructure availability. Bioinformatics analysis was largely undertaken using the supercomputing facilities at CU operated by ARCCA on behalf of the Cardiff Supercomputing Facility and the HPC Wales and Supercomputing Wales projects. The latter is partly funded by the European Regional Development Fund via the Welsh Government. We thank the team of curators for the databases hosted on PubMLST https://pubmlst.org/databases. We thank the curators of the Institut Pasteur MLST and whole-genome MLST databases for curating the *Klebsiella* spp. data and making them publicly available at http://bigsdb.pasteur.fr. We thank M. Islam for providing access to the clinical sites and epidemiology data in Bangladesh. We would like to acknowledge R. Kamran, the microbiologist from PIMS, Pakistan, who sadly passed away in 2018. We thank the team within the Specialist Antimicrobial Chemotherapy Unit, University Hospital Wales, Public Health Wales, for their support for MALDI-TOF MS of bacterial isolates. We thank the BARNARDS group (Supplementary Table 5).

## Author contributions

M.J.C. and K.S. designed and guided the study and analysis, M.N-R. and K.S. wrote the manuscript, and M.N.-R., K.S., and K.M.T. revised the manuscript. K.S., E.A.R.P. and I.B. performed the WGS experiments. M.J.C., K.S., K.M.T., E.A.R.P., J.M., C.D., C.A., P.H., H. Saif, A.D.S.F., M.N.-R., T.H., A.M., A.R., B.P., and L.R. performed the microbiology experiments. K.S., M.J.C. and R.A. performed the bioinformatics analysis. R.M., G.C., C.D. K.M.T., and T.R.W. designed and delivered the epidemiological aspects of the study. W.J.W. performed statistical analyses. R.Z., H. Shirazi., A.M., S.N.U., M.H.J., S.A., K.C.I., F.M., S.U., L.A., C.P.E., A.H.Y., A.A., A.S.M., J.B.M., A.R., L.G., S.M., A.N.H.B., A.W., L.R., D.B., S.S., M.A. and G.M. assisted in collecting hospital surface swab samples and transporting to the United Kingdom. G.C., K.C.I., R.Z., J.B.M. and S.M. facilitated the epidemiology data collection at the clinical sites. T.R.W., M.J.C., R.M., G.C., K.C.I., R.Z., J.B.M. and S.M. designed the BARNARDS study.

## Competing interests

The authors declare no competing interests.

## Additional information

Maria Nieto-Rosado [1,2,25] ✉, Kirsty Sands [1,2,25], Edward A. R. Portal [1,2], Kathryn M. Thomson[1,2], Maria J. Carvalho[2,3], Jordan Mathias[2], Rebecca Milton[2,4], Calie Dyer[2,4], Chinenye Akpulu[1,2], Ian Boostrom[2], Patrick Hogan[2], Habiba Saif[2], Ana D. Sanches Ferreira[2,5], Thomas Hender[2], Barbra Portal[2], Robert Andrews[2], W. John Watkins[2], Rabaab Zahra[6], Haider Shirazi[7], Adil Muhammad [6], Syed Najeeb Ullah[6], Muhammad Hilal Jan[6], Shermeen Akif[6], Kenneth C. Iregbu [8], Fatima Modibbo[9], Stella Uwaezuoke[10], Lamidi Audu [8], Chinago P. Edwin[11,12], Ashiru H. Yusuf[12], Adeola Adeleye[13], Aisha S. Mukkadas[13], Jean Baptiste Mazarati[14], Aniceth Rucogoza[14], Lucie Gaju[14], Shaheen Mehtar[15,16], Andrew N. H. Bulabula[16,17], Andrew Whitelaw [18,19], Lauren Roberts [18], Grace Chan[20,21], Delayehu Bekele[22,23], Semaria Solomon[24], Mahlet Abayneh[21], Gesit Metaferia[24], Group BARNARDS*, Timothy R. Walsh[1,2]

[1]Department of Biology, Ineos Oxford Institute for Antimicrobial Research, University of Oxford, Oxford, UK. [2]Division of Infection and Immunity, Cardiff University, Cardiff, UK. [3]Department of Medical Sciences, Institute of Biomedicine, University of Aveiro, Aveiro, Portugal. [4]Centre for Trials Research, Cardiff University, Cardiff, UK. [5]Parasites and Microbes Programme, Wellcome Sanger Institute Hinxton, Hinxton, UK. [6]Department of Microbiology, Quaid-i-Azam University, Islamabad, Pakistan. [7]Pakistan Institute of Medical Sciences, Islamabad, Pakistan. [8]National Hospital Abuja, Abuja, Nigeria. [9]Débbo Africa, Lekki, Lagos, Nigeria. [10]Federal Medical Centre Jabi, Abuja, Nigeria. [11]Department of Microbiology, Medway Maritime Hospital NHS Foundation Trust, Gillingham, Kent, UK. [12]Aminu Kano Teaching Hospital, Kano, Nigeria. [13]Murtala Muhammad Specialist Hospital, Kano City, Nigeria. [14]The National Reference Laboratory, Rwanda Biomedical Centre, Kigali, Rwanda. [15]Unit of IPC, Stellenbosch University, Cape Town, South Africa. [16]Infection Control Africa Network, Cape Town, South Africa. [17]Department of Global Health, Stellenbosch University, Cape Town, South Africa. [18]Division of Medical Microbiology, Stellenbosch University, Cape Town, South Africa. [19]National Health Laboratory Service, Tygerberg Hospital, Cape Town, South Africa. [20]Department of Pediatrics, Boston Children's Hospital, Harvard Medical School, Boston, MA, USA. [21]Department of Pediatrics and Child Health, St Paul's Hospital Millennium Medical College, Addis Ababa, Ethiopia. [22]Department of Epidemiology, Harvard T.H. Chan School of Public Health, Boston, USA. [23]Department of Obstetrics and Gynecology, St Paul's Hospital Millennium Medical College, Addis Ababa, Ethiopia. [24]Department of Microbiology, Immunology and Parasitology, St Paul's Hospital Millennium Medical College, Addis Ababa, Ethiopia. [25]These authors contributed equally: Maria Nieto-Rosado, Kirsty Sands. *A list of authors and their affiliations appears at the end of the paper. ✉e-mail: maria.nietorosado@biology.ox.ac.uk

## Group BARNARDS

Maria Nieto-Rosado [1,2,25] ✉, Kirsty Sands [1,2,25], Edward A. R. Portal [1,2], Kathryn M. Thomson[1,2], Maria J. Carvalho[2,3], Jordan Mathias[2], Rebecca Milton[2,4], Calie Dyer[2,4], Chinenye Akpulu[1,2], Ian Boostrom[2], Patrick Hogan[2], Habiba Saif[2], Ana D. Sanches Ferreira[2,5], Thomas Hender[2], Barbra Portal[2], Robert Andrews[2], W. John Watkins[2], Rabaab Zahra[6], Haider Shirazi[7], Adil Muhammad [6], Syed Najeeb Ullah[6], Muhammad Hilal Jan[6], Shermeen Akif[6], Kenneth C. Iregbu [8], Fatima Modibbo[9], Stella Uwaezuoke[10], Lamidi Audu [8], Chinago P. Edwin[11,12], Ashiru H. Yusuf[12], Adeola Adeleye[13], Aisha S. Mukkadas[13], Jean Baptiste Mazarati[14], Aniceth Rucogoza[14], Lucie Gaju[14], Shaheen Mehtar[15,16], Andrew N. H. Bulabula[16,17], Andrew Whitelaw [18,19], Lauren Roberts [18], Grace Chan[20,21], Delayehu Bekele[22,23], Semaria Solomon[24], Mahlet Abayneh[21], Gesit Metaferia[24] & Timothy R. Walsh[1,2]

A full list of members and their affiliations appears in the Supplementary Information.

