## [Peer Review File · Nature Communications]

REVIEWER COMMENTS

Reviewer #1 (Remarks to the Author):

This manuscript originates from the Burden of Antibiotic Resistance in Neonates from Developing Societies (BARNARDS) six-country LMIC study. The aim is to determine the prevalence and diversity of antibiotic resistance (ESBL- and carbapenemase)-producing Gram negative bacilli (GNB) recovered from hospital environment and water. Over 6,000 swabs are performed, and ESBL/carbapenemase markers (together depicted as ARG) are screened using Chromagar plates and confirmed using multiplex PCR. The major findings are: 1) high prevalence of ARG-producing GNB from environment, water/sinks and patient care equipment; 2) large difference in the degree of colonized ARG GNB between countries; 3) water is more colonized with ARG-producing GNB than hospital surfaces; 4) temporal variations in ARG prevalence. 128 isolates from 18 different species underwent WGS by Illumina and/or Nanopore; there is evidence of local transmission and links to neonatal sepsis.

The major strength of this study is the multi-national nature (10 hospitals from 6 LMIC countries), the first ever being done. However, there are many weaknesses that dampen the enthusiasm for this project. First, there is lack of consistency of which samples to be swabbed and the number of swabs performed per hospital. Second, water is found to harbor more bacteria with ARG, but when compared between hospitals, the rates of ARG of all sites are combined. Third, all samples from all swabs (including water) are combined together as environment when rate of ARG colonization is calculated (samples include hospital surfaces, patient equipment, G-tubes and Foley catheters, etc). These 3 weaknesses together render comparison of ARG rates between hospitals difficult. Fourth, the genetic relatedness between isolates is poorly described. The number of clinical isolates (denoted as sepsis isolates) are not clearly stated. Although SNP distance is given, a phylogenetic tree comparison between strains might provide a better visualization. Furthermore, since Nanopore sequencing is performed, are the authors able to compare the genetic relationship between plasmids (to suggest horizontal plasmid spread among different species)? Lastly, this is merely a descriptive study, and heavy colonization with ARG GNB has been previously described for LMIC single hospital – thus this study is not original.

Other comments:

Lines 64-66 – sentence needs to be rewritten

Lines 183-188 – are the same items swabbed between different timepoints? Some items might be more colonized than the others

Lines 200-210, Figure 2 and Table S4c – Are different species recovered from different swab sites? In another word, are *Pseudomonas aeruginosa* more prominent from water? Stratifying species per swab sites might be important

Table S7 – difficult to understand 1) abbreviations are not spelled out; 2) what is original versus corrected surfaces? 3) What is N/A? If this reflects not done, why positive or negative PCR result is denoted? 4) Why Foley catheter and G-tube are included in environmental sampling?

Table S3. The total number does not add up (number in individual cell from columns C to H added together is less than number in column I) - See examples in rows 38, 40 etc.

Reviewer #2 (Remarks to the Author):

"Colonisation of extended spectrum β -lactamase- and carbapenemase-producing bacteria on hospital surfaces from low-/middle-income countries" by Nieto-Rosado and colleagues describes a large collection of samples from the hospital environment. 6,290 environmental swabs (from multiple countries and hospitals) were taken and 3,816 of them grew Gram-neg organisms. Of those 13.3% were blaCTX-M-15+, 5.4% were blaNDM+ and 1.2% were bla-OXA48+. Some fraction of those drug resistant organisms were sequenced and clustered into putative transmission clusters with a small collection of neonatal-sepsis isolates. Low SNP counts between isolates (1-10) support transmission across surfaces and, potentially, to patients.

This study will be of interest to clinicians in low- and middle-income countries, where the sampling was carried out, as it reports on important organisms and resistance genes in those countries and their prevalence on surfaces. Healthcare systems in high-income countries, where it can be difficult to get this sort of sampling access, should also take note of the sheer number of surfaces that may harbor cultivatable organisms. The study is a considerable amount of work spanning a number of countries and healthcare systems. The authors have done a nice job of organizing a large amount of data in the supplementary tables.

1. The manuscript is extremely number heavy and, at times, it feels like the authors are simply reading the tables. For instance, in the first paragraph of the results it isn't necessary to show the math for every percentage "n=1,487/6,290 (23.6%)". It is also redundant with Table 1. I would scan the paper for places where large blocks of number-heavy text can be simplified. The paper is very descriptive.
2. There are a lot of p-values throughout the manuscript, including one stuck at the bottom of Table 1 but it was unclear to me what statistical test was being used and what data were being tested (e.g., Table 1). The methods mention Chi-squared test but that's it. Please be more explicit in the text and legends about what statistical tests are being run and how they should be interpreted.
3. Given the importance of the SNP calling, I would like to see more details in the methods instead of just referencing Sands et al and Carvalho et al. It doesn't have to be extensive but a brief summary is needed.

Minor points

- * Title - Consider rewording to: "Colonisation of hospital surfaces from low- /middle-income countries by extended spectrum β -lactamase- and carbapenemase-producing bacteria"
- * Line 102 - HSS define first use
- * Line 48 - throughout, it would be good to make sure you are differentiating sinks, drains, etc... from clean water and taps. "water system" isn't very informative (see also lines 162, 165, 420)
- * Throughout, the authors refer to surfaces as "inanimate" or "inert". I'm not sure what is supposed to be communicated by that. "Fomite" would be the correct technical term. This might be less confusing given that some of the items (e.g., carts/trolleys) while not "animate" are mobile and can potentially be vectors for organisms to travel around. It also doesn't really cover things like various people's "hands" in table S3; are those really inanimate or HSS?
- * Line 83 - "Outbreaks caused by carbapenem-resistant and hypervirulent *K. pneumoniae* have been reported." Sentence needs a reference or something. It seems out of place.
- * Figure 1 legend is too wordy
- * Line 183 - "309 original individual surfaces". What is meant here by "original"
- * Thank you for including the BioProject and genome accessions. They don't appear to be accessible/released but I assume that they will be on publication.

RESPONSE TO THE REVIEWERS' COMMENTS

Legend:

Black text: comments from reviewers

Blue text: author's response to reviewers' comments for resubmission.

Dear reviewer's,

We would like to thank you for the comments and queries raised during the review of our manuscript, and for allowing a re-submission. We have carefully considered these during our revision and provide a point-by-point response below.

Reviewer #1 (Remarks to the Author):

This manuscript originates from the Burden of Antibiotic Resistance in Neonates from Developing Societies (BARNARDS) six-country LMIC study. The aim is to determine the prevalence and diversity of antibiotic resistance (ESBL- and carbapenemase)-producing Gram negative bacilli (GNB) recovered from hospital environment and water. Over 6,000 swabs are performed, and ESBL/carbapenemase markers (together depicted as ARG) are screened using Chromagar plates and confirmed using multiplex PCR. The major findings are: 1) high prevalence of ARG-producing GNB from environment, water/sinks and patient care equipment; 2) large difference in the degree of colonized ARG GNB between countries; 3) water is more colonized with ARG-producing GNB than hospital surfaces; 4) temporal variations in ARG prevalence. 128 isolates from 18 different species underwent WGS by Illumina and/or Nanopore; there is evidence of local transmission and links to neonatal sepsis.

The major strength of this study is the multi-national nature (10 hospitals from 6 LMIC countries), the first ever being done. However, there are many weaknesses that dampen the enthusiasm for this project.

1. First, there is lack of consistency of which samples to be swabbed and the number of swabs performed per hospital.

We do agree with the reviewer here, and have pointed this out as a limitation of the study as it was extremely difficult to standardise swabbing on occasion during the study, and there was a variation in the outlay of the hospital wards throughout the BARNARDS network. Accordingly, we were cautious with our interpretation of the data and we therefore did not study correlation between Gram-negative bacteria (GNB) growth and/or antibiotic resistance gene (ARG) prevalence per surface type or per each of the hospital sites. We do plan to address this limitation in the next phase of BARNARDS, which will commence towards the end of 2023.

2. Second, water is found to harbour more bacteria with ARG, but when compared between hospitals, the rates of ARG of all sites are combined.

We apologise for the confusion regarding the "water samples". We collected charcoal swabs with a label indicating the swab originated near a water source but no further information was provided (e.g., samples labelled as "tap water", and therefore it was classified under the category sinks and water system). As the term for the category leads to confusion, we have amended the text accordingly to clarify this category (please refer to a similar comment from reviewer 2, comment 15). We agree that the correlation between GNB growth and/or ARG prevalence per each surface type and per each country/hospital would have been a preferred analysis to observe whether certain surfaces are more contaminated in a hospital. Unfortunately, the limitations within the study (sampling consistency) prevented analysis on the country/site level in this case and therefore we have combined these data. Following categorisation into six surface categories (600-700 sample average per category, Supplementary Table S10 [S8 in submission]), our analysis permitted us to determine which surfaces were colonised with ARG carrying-bacteria but not per

hospital site. We appreciate this comment, and have accordingly created an additional Supplementary Table (new S5) presenting ARG rates per surface category, and also per hospital site and country. We are aware that the results are not significant because of the sample number limitation per site, and we have clearly stated this in the article. The text has been amended accordingly to incorporate this into, the limitation section in the discussion.

3. Third, all samples from all swabs (including water) are combined together as environment when rate of ARG colonization is calculated (samples include hospital surfaces, patient equipment, G-tubes and Foley catheters, etc).

We appreciate the umbrella use of the term “environment” might be confusing; however, we have considered all hospital surfaces as "environmental" in the hospital/clinical context for this manuscript. Please, see comment 8d). In relation to a previous comment, we have added descriptive details and frequencies regarding ARG colonisation per each surface category (new S5); and Supplementary Table S4 shows ARG per surface type.

These 3 weaknesses together render comparison of ARG rates between hospitals difficult.

We agree with this summary and these sampling limitations have prevented detailed hospital/country-level analysis. Nevertheless, this is a very large dataset that will add considerable emphasis to otherwise scant data on the topic of infection prevention and control (IPC). We hope that this data will bring some much needed attention on IPC in many LMIC sites (a key topic tabled for the UN AMR high level meeting in 2024), regarding the amount of GNB growth found in hospitals and associated ARGs. We hope that this dataset will emphasise that the importance of cleaning hospital surfaces can aid in informing future IPC practices. We are looking forward to building on this study during our next phase of the BARNARDS project, with more standardised sampling allowing prevalence to be assessed per hospital site..

4. Fourth, the genetic relatedness between isolates is poorly described.
 - a. The number of clinical isolates (denoted as sepsis isolates) are not clearly stated.

We thank the reviewer for highlighting missing information and linkage to our previous publication (reference now added). There were a total number of 22 ST 15 *K. pneumoniae* causing sepsis in Pakistan characterised in BARNARDS¹. This has been added for resubmission in Results section 2.6.

- b. Although SNP distance is given, a phylogenetic tree comparison between strains might provide a better visualization.

We appreciate that phylogenetic trees allow better visualisation of larger clusters, particularly ST15 *K. pneumoniae*. However, we have opted to include a figure that summarises multiple STs/bacterial isolates (Figure 4 for resubmission, submitted as Figure 3) instead of creating phylogenetic trees for this article. Please also note that many clusters are quite small, which would limit the impact of a phylogenetic tree, and, for ST15 *K. pneumoniae*, there is an additional dataset linked to samples collected from the mother and neonatal rectal samples (DOI: [10.1038/s41564-022-01184-y](https://doi.org/10.1038/s41564-022-01184-y)) that will be soon submitted for publication elsewhere and includes larger phylogenetic trees.

- c. Furthermore, since Nanopore sequencing is performed, are the authors able to compare the genetic relationship between plasmids (to suggest horizontal plasmid spread among different species)?

We thank the reviewer for this suggestion - we have performed additional analysis to incorporate plasmid analysis and genetic relationships between plasmids where appropriate, using available clinical and metadata to suggest whether there is evidence to support horizontal plasmid transfer. Please see section 2.5 and the addition of a new main figure, Figure 3 – a Sankey diagram detailing information regarding linkage between carbapenemase gene variants and plasmid incompatibility groups. Furthermore, a comprehensive analysis of plasmid sequences (assemblies generated during whole genome assembly as outlined in the methods) has been summarised as a separate document called Supplementary Dataset 1.

Lastly, this is merely a descriptive study, and heavy colonization with ARG GNB has been previously described for LMIC single hospital – thus this study is not original.

We acknowledge that the bulk of the data presented within this manuscript is descriptive. However, further to a comprehensive literature search, there are no other studies undertaken on multinational (13 sites and 8 counties) scale linking bacterial colonisation on hospital surfaces of AMR bacteria correlating with neonatal sepsis. We believe this study showcases the global issues and prevalence of ARGs. An important benefit of this study is the unified microbiology culture of samples from each hospital which allows aggregated data analysis across the BARNARDS network. The additional genomics analysis (in response to your earlier comment) further provides evidenced novelty within this manuscript. We highlight a large diversity of strains, and whilst the microbiology data alone is not novel, the addition of long-read sequencing on this scale has allowed us to compare plasmids and identify cases of both clonal spread of bacterial strains across hospital surfaces, as well as cases of potential plasmid transfer between bacterial species. We would like to thank the reviewer for their comment in relation to the plasmid analysis, as we believe this has improved the quality of our analysis. In totality, our data summarises clear evidence that access to IPC measures is essential to limit transmission of AMR bacteria. We have added and edited the discussion around the impact of our data to lines 348-351 from the revised manuscript²⁻⁹.

5. Other comments: Lines 64-66 – sentence needs to be rewritten

We thank the reviewer for this feedback, we have now amended these lines.

6. Lines 183-188 – are the same items swabbed between different timepoints? Some items might be more colonized than the others

We thank the reviewer for raising this point. This analysis is a pooled data, and whilst some items are swabbed at different timepoints, this was not strictly specified during sampling. We agree we cannot assume there is a correlation between ARG prevalence and certain time points. Bacterial colonisation likely depends on multiple factors including the type of surface and, our sampling limitations prevents answering this query more fully. However, we do state that ARG-carrying bacteria are more frequently detected in HSS collected between March and October (lines 367-369 in submitted manuscript); and consequently, we have acknowledged that “*Due to inconsistent sampling among time periods, seasonal analysis per country was not performed.*” We have amended lines 187-188 in the submitted manuscript (lines 176-177 in the revised version). We agree with the hypothesis that highly contaminated surfaces may have been swabbed more often during these time episodes. Our aim, with this analysis, was to highlight certain patterns of clonal prevalence and state any observed differences between time bands throughout the study period. We did not aim to specifically correlate colonisation to months of the year, and our analysis is exploratory in nature. We considered existing differences

not only in type of surfaces collected, but also in seasons between countries and even areas within the same country (e.g. seasons in Cape Town and the rest of South Africa are different). We did not make prevalence comparisons between seasons or times of the year, and per hospital/country. We have amended the text, e.g. submitted lines 368-369, and limitations (linked to comment 2).

7. Lines 200-210, Figure 2 and Table S4c – Are different species recovered from different swab sites? In another word, are *Pseudomonas aeruginosa* more prominent from water? Stratifying species per swab sites might be important

We agree with the reviewer's suggestion and carbapenemase-carrying bacterial species are stratified per hospital site in Supplementary Table S6 (submitted as S4), so the reader can observe the bacterial species recovered in each hospital site. Because of our main limitation with the swabbing, we cannot state that a specific bacterial species was more frequently found in one hospital site or surface category (explanation in comment 1-3). Moreover, in this study we screened for the presence of ESBL or carbapenemase ARGs within the bacterial species colonising the hospital surfaces rather than delineating the bacterial profile. Due to our approach, we are unable to accurately determine whether *P. aeruginosa* or any other bacterial species is more frequently found in a specific surface within a hospital, and that is fundamentally why Supplementary Table S6 shows bacterial species recovered per hospital site and not per surface category. Additionally, this is the reason why surface category was considered for potential transmission conclusion (Figure 4) and not for correlation with bacterial species. In the discussion section (lines 351-354 and lines 356-358 from the submitted manuscript), we did not make any correlation between bacterial species and surface type or category. Therefore, we have stated that “recovery from HSS”, “surfaces in the ward”, or “multiple clusters detected from hospital surfaces”. We agree that analysing between bacterial species versus surface category is important when considering environmental reservoirs in the hospital settings to understand if certain surfaces pose a higher threat to patients/outbreaks if a particular species is found in a particular environment. Supplementary Table S7 has been created for this resubmission, which shows n (%) of bacterial species and isolates carrying *bla_{NDM}* and *bla_{OXA-48}*-like per surface category. We anticipate that our next phase of BARNARDS will fully address this research question.

8. Table S7 – difficult to understand.

We thank the reviewer for this feedback and have modified the table accordingly.

- a. 1) abbreviations are not spelled out; these are now all added in Supplementary Table S10 (submitted as S7)
- b. 2) what is original versus corrected surfaces? This has been addressed and identified by both reviewers, please see a detailed response in comment 19 from reviewer 2.
- c. 3) What is N/A? If this reflects not done, why positive or negative PCR result is denoted? N/A means PCR was not performed for those samples with no growth (N) on VC (NA for *bla_{CTX-M-15}* screening) or for VE (NA for carbapenemase genes).
- d. 4) Why are Foley catheter and G-tube included in environmental sampling?

We have added text for clarity in the methods, and results/discussions as appropriate (please also see comments 15 and 16 from reviewer 2).

Gastrostomy tubes or “feeding tubes” (as reported by the hospital sites), as well as foley catheter, were included within the medical equipment category. According to the first definition used in our methods section, we collected hospital surface swabs (HSS). However, the terms “environmental surfaces and patient care equipment” have been used in the discussion, following CDC guideline definitions¹⁰. In methods we used the terms “medical

equipment and inanimate surfaces”, following WHO definitions^{5,8,11} considered to be equally used for surfaces surrounding the patient immediate surroundings. The use of “inanimate” has been eliminated from the paper. Terms will be used according to CDC guidelines¹⁰ throughout the paper: environmental surfaces (e.g., bed, cot, bed rails, mattresses) and patient care equipment (also described in this work as medical equipment or mobile medical equipment, including non-critical, semi-critical and noncritical patient care equipment) have been clarified in the methods section. Creating the two categories “medical equipment” and “mobile medical equipment” allowed us to distinguish between patient care equipment dedicated to a patient (direct patient care), and equipment moving across wards used for different patients. To conclude, feeding tubes and foley catheter would have been included within critical patient care equipment, as these devices enter sterile tissue or the vascular system. Being aware of this, these are classified under the umbrella of “patient care equipment” or, as defined in this manuscript, “medical equipment”.

9. Table S3. The total number does not add up (number in individual cell from columns C to H added together is less than number in column I) - See examples in rows 38, 40 etc.

We thank the reviewer for noting the differences in numbers and we understand the difficulty with the number changes throughout the manuscript. As outlined in the submitted Table S3 (S4 in resubmission) caption “The total number of samples collected per individual surface (over 4,126) was used as denominator to calculate the prevalence of each ARG per each individual surface”. Also, the second column (column I in the submitted S3) shows the number of samples collected per type of surface (total of 4,126). This column might be not adding up with the ARGs counts since more than one PCR reaction was performed on the 4,126 samples (samples might have been positive for more than one ARG). Table and caption have been amended to avoid confusion.

Reviewer #2 (Remarks to the Author):

"Colonisation of extended spectrum β -lactamase- and carbapenemase-producing bacteria on hospital surfaces from low-/middle-income countries" by Nieto-Rosado and colleagues describes a large collection of samples from the hospital environment. 6,290 environmental swabs (from multiple countries and hospitals) were taken and 3,816 of them grew Gram-neg organisms. Of those 13.3% were blaCTX-M-15+, 5.4% were blaNDM+ and 1.2% were bla-OXA48+. Some fraction of those drug resistant organisms were sequenced and clustered into putative transmission clusters with a small collection of neonatal-sepsis isolates. Low SNP counts between isolates (1-10) support transmission across surfaces and, potentially, to patients.

This study will be of interest to clinicians in low- and middle-income countries, where the sampling was carried out, as it reports on important organisms and resistance genes in those countries and their prevalence on surfaces. Healthcare systems in high-income countries, where it can be difficult to get this sort of sampling access, should also take note of the sheer number of surfaces that may harbour cultivatable organisms. The study is a considerable amount of work spanning a number of countries and healthcare systems. The authors have done a nice job of organizing a large amount of data in the supplementary tables.

10. The manuscript is extremely number heavy and, at times, it feels like the authors are simply reading the tables. For instance, in the first paragraph of the results it isn't necessary to show the math for every percentage "n=1,487/6,290 (23.6%)". It is also redundant with Table 1. I would scan the paper for places where large blocks of number-heavy text can be simplified. The paper is very descriptive.

We thank the reviewer for highlighting how number intensive the results section is and the manuscript text has been reduced and edited as suggested during this revision, including the removal of numbers that replicate data presented in Table 1.

11. There are a lot of p-values throughout the manuscript, including one stuck at the bottom of Table 1 but it was unclear to me what statistical test was being used and what data were being tested (e.g., Table 1). The methods mention Chi-squared test but that's it. Please be more explicit in the text and legends about what statistical tests are being run and how they should be interpreted.

We thank the reviewer for highlighting a shortage of information about statistical tests and results. The methods section has been amended (line 546-552 for submission, 526-531 for revised version) and we have edited Table 1, 2 and 3 legends (Tables 1,3,4 for resubmission). P-values throughout the manuscript have been complemented with more details regarding the test and variables comparison performed. For instance, line 109-113 in submitted manuscript.

12. Given the importance of the SNP calling, I would like to see more details in the methods instead of just referencing Sands et al and Carvalho et al. It doesn't have to be extensive but a brief summary is needed.

We agree and apologise for the oversight and have added some details.

Minor points

13. * Title - Consider rewording to: "Colonisation of hospital surfaces from low- /middle-income countries by extended spectrum β -lactamase- and carbapenemase-producing bacteria"

We appreciate the suggestion and we have changed the title accordingly.

14. * Line 102 - HSS define first use

We have checked for the first use of the abbreviations during revision.

15. * Line 48 - throughout, it would be good to make sure you are differentiating sinks, drains, etc... from clean water and taps. "water system" isn't very informative (see also lines 162, 165, 420)

"Sinks and water system" has been changed to "surfaces near the sink drain". This term has been described in methods (together with 8d) and includes sink basin, faucet, faucet handles, and surrounding countertop. This term has been adopted from CDC reported evidence of sinks and other drains which may easily become contaminated with multidrug-resistant organisms (MDRO)^{12,13}, which can persist for a long period and are often difficult to remove, especially when IPC practices are limited^{12,13}. Reducing healthcare-associated infections (HAIs) is included in the WHO action plan¹⁴, and CDC recommends reducing risk from water (including the term surfaces near the drain) as healthcare environmental infection prevention measure, and in this manner to prevent HAIs^{12,13}. Due to limited information on sampling, HSS collected from surfaces near the sink drain were classified together.

16. * Throughout, the authors refer to surfaces as "inanimate" or "inert". I'm not sure what is supposed to be communicated by that. "Fomite" would be the correct technical term. This might be less confusing given that some of the items (e.g., carts/trolleys) while not "animate" are mobile and can potentially be vectors for organisms to travel around. It also doesn't really cover things like various people's "hands" in table S3; are those really inanimate or HSS?

We thank the reviewer for raising this concern. We have eliminated the use of inanimate throughout the manuscript (see comment 8d). In relation to healthcare workers hands (e.g.

baby guardian, clinician, doctor, nurse, cleaner), as well as mother hand, these samples have been considered environmental surfaces and have been classified in patient zone. Surfaces in the patient zone are contaminated by bacteria colonising/infecting patients^{7-11,15} in two ways: direct shedding from patients and via healthcare workers hands¹⁶. According to CDC guideline best practices, the role of environmental surfaces, environmental cleaning but also hand hygiene play a role in the contact transmission pathway and in breaking the chain of this transmission. Contaminated hands of healthcare personnel can contaminate environmental surfaces unless proper hand hygiene and environmental cleaning are properly performed to prevent MDRO transmission to susceptible patients or their immediate surrounding. According to the Hand Hygiene Technical Reference Manual¹⁷, it is possible to prevent HAIs caused by cross-transmission via hands if hand hygiene is correctly followed. At the same time, cross-transmission could potentially happen by contamination of healthcare personnel from hand contact with contaminated environmental surfaces, patient care equipment, or patients¹⁸. Thus, healthcare workers hands, as well as mother hands, can act as reservoirs and could potentially contribute to cross-transmission when providing direct and indirect care. To avoid confusion for the reader, this point has been clarified in methods, together with other terms amended (comments 8d) and 15).

17. * Line 83 - "Outbreaks caused by carbapenem-resistant and hypervirulent *K. pneumoniae* have been reported." Sentence needs a reference or something. It seems out of place.
We thank the reviewer for this feedback. We have now amended the text and added references. This text has been moved to discussion (line 344).

18. * Figure 1 legend is too wordy
We thank the reviewer for this observation. Figure 1 caption has been reduced.

19. * Line 183 – “309 original individual surfaces”. What is meant here by “original”
The use of “original” in this way refers to the surface label provided before extended data cleaning and aggregation into categories. We used “original” to refer to the 309 different HSS collected “initially” by the hospital sites, and to differentiate them from the categories created. We have now amended the text for clarity and have eliminated the terms “original” and “individual” (line 183 in submitted manuscript, Supplementary Table S10 (submitted as S7) and Supplementary Table S4).

* Thank you for including the BioProject and genome accessions. They don't appear to be accessible/released but I assume that they will be on publication.
We thank the reviewer for highlighting this, all data is now in the public domain.

ADDITIONAL COMMENTS FROM AUTHORS:

- After reanalysis in section 2.5 (comment 4, reviewer 1) we identified that the total number of isolates carrying *bla*_{OXA-48}-like was 17 and not 18. This was likely an Error in Microsoft Excel sorting and has now been corrected.
- As new figures and supplementary material was created, the supplementary information has also been amended.
- During revision it was necessary to add new references. All new references are cited below for information:

1. Sands, K. *et al.* Characterization of antimicrobial-resistant Gram-negative bacteria that cause neonatal sepsis in seven low- and middle-income countries. *Nat. Microbiol.* **6** (2021), 512–523, (2021).
2. World Health Organization. *Global report on the epidemiology and burden of sepsis.* <https://www.who.int/publications-detail-redirect/9789240010789> (2020).
3. World Health Organization. Global progress report on WASH in health care facilities: Fundamentals first. <https://www.who.int/publications-detail-redirect/9789240017542> (2021).
4. World Health Organization. Regional Office for the Western Pacific. *Practical guidelines for infection control in health care facilities.* (2004).
5. World Health Organization. *Guidelines on core components of infection prevention and control programmes at the national and acute health care facility level.* (2016).
6. World Health Organization. Global report on infection prevention and control. <https://www.who.int/publications-detail-redirect/9789240051164> (2022).
7. World Health Organization. *Guidelines for the prevention and control of carbapenem-resistant Enterobacteriaceae, Acinetobacter baumannii and Pseudomonas aeruginosa in health care facilities.* (2017).
8. World Health Organization. *Minimum requirements for infection prevention and control programmes.* (World Health Organization, 2019).
9. World Health Organization. *Report on the burden of endemic health care-associated infection worldwide.* (2011).
10. CDC and ICAN. *Best practices for environmental cleaning in resource-limited healthcare settings. A healthcare cleaning and disinfection guide for healthcare settings with limited resources.* <https://www.cdc.gov/hai/prevent/resource-limited/index.html> (2023).

11. World Health Organization & WHO Patient Safety. WHO guidelines on hand hygiene in health care. (2009).
12. Centers for Disease Control and Prevention (CDC). Public Health Strategies to Prevent the Spread of Novel and Targeted Multidrug-resistant Organisms (MDROs). (2023).
13. Centers for Disease Control and Prevention (CDC). Preventing Healthcare-Associated Infections. Reduce Risk from Water.
<https://www.cdc.gov/hai/prevent/environment/water.html> (2023).
14. World Health Organization. Global strategy on infection prevention and control.
<https://www.who.int/publications/m/item/global-strategy-on-infection-prevention-and-control>.
15. World Health Organization. *Decontamination and reprocessing of medical devices for health-care facilities*. (World Health Organization, 2016).
16. Russotto, V. *et al.* What Healthcare Workers Should Know about Environmental Bacterial Contamination in the Intensive Care Unit. *BioMed Res. Int.* **2017 (6905450)**, 7, (2017).
17. World Health Organization & WHO Patient Safety. Hand hygiene technical reference manual: to be used by health-care workers, trainers and observers of hand hygiene practices. (2009).
18. CDC. Guideline for Disinfection and Sterilization in Healthcare Facilities, 2008. (2019).

REVIEWERS' COMMENTS

Reviewer #1 (Remarks to the Author):

The authors have adequately addressed the majority of concerns raised by the reviewers. The limitations of the data are adequately acknowledged. The reviewer appreciates the significant revision which renders the manuscript better.

Recommendations:

- 1) Involve a native English speaker to assist with the manuscript final readthrough
- 2) The results section is still “number heavy”. Since there are many tables (both in main text and in the supplementary materials), would highlight the major points the authors try to make, summarize the data and just refer to the tables/figures.
- 3) Few specific critiques:
 - a. Abstract – Line 46 should be were rather than “was”
 - b. Introduction - Last sentence should belong in Methods
 - c. Results
 - i. Table 1 – line 101. Not sure what “some or all the proportion” means. In addition, Table 1 does not show type of surfaces, so why p-value was given for this variable (line 100)?
 - ii. Are Barnards hospitals the same for Tables 1, 2 and Fig 1. If yes, then why the description of BARNARDS hospitals are repeated?
 - iii. Line 151. What is the p-value for the HSS+ for CTX-M between medical equipment and surfaces to render the difference “significant?”. Same comment for statements in line 153-156
 - iv. Table 3. P-values for what comparisons are not well-defined
 - v. Table 4. Is chi-square an appropriate test to evaluate difference in percentage of specific group over times?

POINT-BY-POINT RESPONSE TO REVIEWERS' COMMENTS

Legend:

Black text: comments from reviewers

Blue text: author's response to reviewers' comments for resubmission.

Dear reviewers,

We would like to thank you for the comments and queries raised during the review of our manuscript, and for allowing a re-submission. We have carefully considered these during our revision and provide a point-by-point response below.

Reviewer #1 (Remarks to the Author):

The authors have adequately addressed the majority of concerns raised by the reviewers. The limitations of the data are adequately acknowledged. The reviewer appreciates the significant revision which renders the manuscript better.

Recommendations:

- 1) Involve a native English speaker to assist with the manuscript final readthrough
- 2) The results section is still "number heavy". Since there are many tables (both in main text and in the supplementary materials), would highlight the major points the authors try to make, summarize the data and just refer to the tables/figures.

We thank the reviewer for the suggestions. We agree with recommendation 2 and we have revised the manuscript accordingly.

3) Few specific critiques: We thank the reviewer for raising these points. We do agree with the critiques and have amended the manuscript accordingly:

a. Abstract – Line 46 should be were rather than "was" The tense in the abstract has been amended.

b. Introduction - Last sentence should belong in Methods

c. Results

- i. Table 1 – line 101. Not sure what "some or all the proportion" means. In addition, Table 1 does not show type of surfaces, so why p-value was given for this variable (line 100)?
- ii. Are Barnards hospitals the same for Tables 1, 2 and Fig 1. If yes, then why the description of BARNARDS hospitals are repeated?
- iii. Line 151. What is the p-value for the HSS+ for CTX-M between medical equipment and surfaces to render the difference "significant?"). Same comment for statements in line 153-156
- iv. Table 3. P-values for what comparisons are not well-defined

A single chi-squared test was performed, an omnibus test which determines whether there are or not differences overall (between all surface categories); therefore, P values were obtained when comparing prevalence of each ARG across different surface categories. We could provide the reader with pairwise comparisons, but it will add more text to the manuscript we are attempting to reduce. Table 3 caption has been amended.

v. Table 4. Is chi-square an appropriate test to evaluate difference in percentage of specific group over times?

Since we had count in groups, a contingency table was created. We studied independence of ARG or GNB colonisation (variable 1) over time band (variable 2), to see which combination of variables was over or underrepresented (significant differences between ARG/GNB over time bands). Each of the variables are counted only once, and we are working with different variables, which makes it an omnibus test.

As discussed in the manuscript, inconsistent sampling throughout the hospital sites among time periods (different sample size per month but also per country, and country climate) did only allow us to study ARG/GNB detection differences. That is why we considered Chi-square the most appropriate test, and we studied proportions (frequencies) of ARG or GNB recovered during each time band.